

# Stable isotope analysis ($\delta^{13}$C and $\delta^{15}$N) of soil nematodes from four feeding groups

Carol Melody[1,2], Bryan Griffiths[3], Jens Dyckmans[4] and Olaf Schmidt[1,2]

[1] School of Agriculture and Food Science, University College Dublin, Dublin, Ireland
[2] UCD Earth Institute, University College Dublin, Dublin, Ireland
[3] Crop and Soil Systems Research, Scotland's Rural College, Edinburgh, United Kingdom
[4] Centre for Stable Isotope Research and Analysis, Georg-August Universität Göttingen, Göttingen, Germany

Corresponding author
Carol Melody,
carol.melody@ucdconnect.ie

## ABSTRACT

Soil nematode feeding groups are a long-established trophic categorisation largely based on morphology and are used in ecological indices to monitor and analyse the biological state of soils. Stable isotope ratio analysis ($^{13}$C/$^{12}$C and $^{15}$N/$^{14}$N, expressed as $\delta^{13}$C and $\delta^{15}$N) has provided verification of, and novel insights into, the feeding ecology of soil animals such as earthworms and mites. However, isotopic studies of soil nematodes have been limited to date as conventional stable isotope ratio analysis needs impractically large numbers of nematodes (up to 1,000) to achieve required minimum sample weights (typically >100 μg C and N). Here, micro-sample near-conventional elemental analysis–isotopic ratio mass spectrometry ($\mu$EA–IRMS) of C and N using microgram samples (typically 20 μg dry weight), was employed to compare the trophic position of selected soil nematode taxa from four feeding groups: predators (*Anatonchus* and *Mononchus*), bacterial feeders (*Plectus* and *Rhabditis*), omnivores (Aporcelaimidae and Qudsianematidae) and plant feeder (*Rotylenchus*). Free-living nematodes were collected from conventionally and organically managed arable soils. As few as 15 nematodes, for omnivores and predators, were sufficient to reach the 20 μg dry weight target. There was no significant difference in $\delta^{15}$N ($p = 0.290$) or $\delta^{13}$C ($p = 0.706$) between conventional and organic agronomic treatments but, within treatments, there was a significant difference in N and C stable isotope ratios between the plant feeder, *Rotylenchus* ($\delta^{15}$N = 1.08 to 3.22 mUr‰, $\delta^{13}$C = −29.58 to −27.87 mUr) and all other groups. There was an average difference of 9.62 mUr in $\delta^{15}$N between the plant feeder and the predator group ($\delta^{15}$N = 9.89 to 12.79 mUr, $\delta^{13}$C = −27.04 to −25.51 mUr). Isotopic niche widths were calculated as Bayesian derived standard ellipse areas and were smallest for the plant feeder (1.37 mUr$^2$) and the predators (1.73 mUr$^2$), but largest for omnivores (3.83 mUr$^2$). These data may reflect more preferential feeding by the plant feeder and predators, as assumed by classical morphology-based feeding groups, and indicate that omnivory may be more widespread across detritivore groups i.e. bacterial feeders (3.81 mUr$^2$). Trophic information for soil nematodes derived from stable isotope analysis, scaled as finely as species level in some cases, will complement existing indices for soil biological assessment and monitoring, and can potentially be used to identify new trophic interactions in soils. The isotopic technique used here, to compare nematode feeding group members largely confirm their trophic relations based on morphological studies.

## INTRODUCTION

Nematodes are an abundant and diverse animal group in most soils, especially where decomposition is active (*Bongers & Bongers, 1998*). Nematodes play major roles in soil processes, both directly and indirectly through elemental cycling and decomposition of organic matter. For example, they mineralise nitrogen and phosphorus, as well as influence other soil organisms involved in nutrient cycling (*Ferris et al., 2012*), especially by regulating soil microbial populations (*Griffiths, 1990*). Some soil nematodes feed directly on plants and many are prey for larger soil fauna (*Curry & Schmidt, 2007*; *Heidemann et al., 2011*).

Soil nematodes are traditionally assigned to feeding groups according to morphology, feeding experiments and gut content analyses (*Overgaard-Nielsen, 1949*; *Wood, 1973*; *Yeates et al., 1993*). Nematode feeding groups, functional guilds and strategy-based indices have been used extensively to document the response of nematodes to soil disturbance as bio-indicators of general biological conditions in soil ecosystems (*Neher, 2001*; *Ferris, Bongers & De Goede, 2001*; *Ferris et al., 2012*), and, in ecological studies, to assess the importance of nematodes in soil energy pathways (*De Ruiter, Neutel & Moore, 1998*; *Zhao & Neher, 2014*). The indices developed for soil nematodes have been shown to be applicable to other soil fauna (*Sánchez-Moreno et al., 2009*).

There are, however, discontinuities and uncertainties in the assumed trophic groups of some nematodes. For example, bacterial feeders have been cultured successfully on contrary food sources such as fungi, in laboratory situations, and it is often difficult to assign feeding types at a species level (*Yeates et al., 1993*; *Ferris, Bongers & De Goede, 2001*). Laboratory-based feeding experiments are not always indicative of natural *in situ* feeding behaviour and, morphology alone may be misleading.

Terrestrial and aquatic nematode feeding can be categorised similarly (*Moens, Traunspurger & Bergtold, 2006*) with growing support for a collective classification (*Moens, Yeates & De Ley, 2004*). Feeding response of nematode trophic groups may not be represented fully, without testing finer resolution taxonomic groups (*Neher & Weicht, 2013*; *Cesarz et al., 2013*) and certain groups (i.e., omnivores) may shift trophic level feeding as a result of life stage development (*Moens, Traunspurger & Bergtold, 2006*). Omnivorous nematodes are taken as generalist feeders and less so as 'true' omnivores (*Moens, Yeates & De Ley, 2004*), however, 'true' omnivory (i.e., feeding across different trophic levels) may be more widespread than once assumed in soil food webs (*Scheu, 2002*), and nematode communities are no exception to this theory (*Moens, Traunspurger & Bergtold, 2006*). Several experts have identified the confirmation of trophic groupings of nematodes as a major gap in free-living nematode research (*Scheu, 2002*; *Neher, 2010*; *Ferris et al., 2012*).

In current soil food web studies, the combination of traditional taxonomic and observational techniques with molecular and isotopic advances is yielding novel insights (e.g., *Curry & Schmidt, 2007*). For trophic studies, stable isotopes provide different, often complementary information to molecular techniques because diet-indicating isotopes are assimilated and hence detectable over longer time spans than ingested nucleic acids of food items (*Darby & Neher, 2012*).

To date, isotopic studies have been applied more to aquatic nematode groups than to soil groups and mostly to taxa of larger sizes that yield sufficient sample mass for analysis. For example, in estuarine sediments, C and N isotope measurements showed distinct trophic groupings often coinciding with mouth morphology, but certain assumed deposit feeding taxa without teeth had elevated $^{15}N/^{14}N$ ratios suggesting predatory behaviour (*Moens, Bouillon & Gallucci, 2005*; *Vafeiadou et al., 2014*). Another example is food selectivity of aquatic, bacteria-feeding nematodes, which were investigated by *Estifanos, Traunspurger & Peters (2013)* using isotopically-labelled bacteria, with results suggesting a significant component of algae and diatoms in the diet. Results conflicted so much for *Vafeiadou et al. (2014)* that they concluded that interpretation of nematode feeding ecology based purely on mouth morphology should be avoided.

Soil food webs were traditionally defined with a $\delta^{15}N$ gap of 3.4 mUr (‰) between trophic levels (*Ponsard & Arditi, 2000*). For soil nematodes, plant-parasitic Longidoridae, were first analysed isotopically at species level by *Neilson & Brown (1999)*, and showed varied $\delta^{15}N$ shifts after 28 days on *Petunia* sp. roots when transferred from an isotopically distant host plant, suggesting either different species feeding, metabolism or reproductive mechanisms. Soil food web studies under controlled conditions have analysed entire nematode communities for isotopic comparisons with other fauna groups (*Sampedro & Domínguez, 2008*; *Crotty et al., 2014*), but individual soil nematode trophic group studies have been slow to follow. For instance, the energy channel (whether fungal or bacterial) and $^{13}C$ of soil nematode feeding groups was altered by experimentally raised $CO_2$ with depleted $\delta^{13}C$ ($\approx -47$ mUr), under different crops, in a study by *Sticht et al. (2009)*. In combination with $^{15}N$ analysis, fatty acids compositions were used as traceable markers for trophic studies by *Ruess et al. (2004)*, and the same approach was employed later to show trophic links with $^{13}C$ analysis of individual fatty acids for consumer and predatory soil fauna diets under organic compared with conventional systems (*Haubert et al., 2009*). While these examples enlighten aspects of nematode feeding and its contribution to the larger soil food web, testing of morphology-based nematode feeding group classification has not been extensively undertaken.

Coming closer to this undertaking, *Shaw et al. (2016)* used $^{13}C$ labelled roots to highlight the role of higher trophic level nematodes in soil C flow and root decomposition under burnt prairie grass in a greenhouse experiment. Most recently, using conventional isotopic ratio mass spectrometry (IRMS), a study in a boreal forest showed that soil nematodes from four feeding groups had distinct isotopic values ($\delta^{13}C$ and $\delta^{15}N$) at natural abundance level, representing chiefly trophic differences between microbial and predatory feeders (*Kudrin, Tsurikov & Tiunov, 2015*). Isotopic analysis of soil nematodes using conventional IRMS has been limited by the amount of tissue required to measure N and C (*Darby & Neher, 2012*). Recently, *Langel & Dyckmans (2014)* developed a $\mu$EA–IRMS method that analyses microgram samples (as little as 0.6 $\mu$g for $^{15}N$ and 1 $\mu$g for $^{13}C$). This method has already been used to investigate resource shifts ($^{13}C$ labelled) in soil mesofauna under fertilizer treatments (*Lemanski & Scheu, 2014*) and the comparative feeding ecology of oribatid mites in varying regional and forest deadwood types (*Bluhm, Scheu & Maraun, 2015*).

Here, the $\mu$EA–IRMS method was employed for natural abundance, dual stable isotope analysis of feeding group members of free-living soil nematodes collected from a field experiment with conventionally and organically managed arable soil. This pilot study had three main aims: (i) to establish how many nematodes are needed (from different taxa/groups) for sufficient sample mass for natural abundance isotopic analysis (dual $^{13}$C and $^{15}$N analysis); (ii) to compare members of nematode feeding groups from two different agronomic systems; and (iii) to compare isotopically derived functional group results with traditional nematode feeding classifications.

Isotopic 'niche spaces' were calculated for: predators (*Anatonchus* and *Mononchus*), bacterial feeders (*Plectus* and *Rhabditis*), omnivores (Aporcelaimidae and Qudsianematidae) and the plant feeder (*Rotylenchus*). We hypothesized that (1) the isotopically represented nematode communities would be altered under the organically amended agronomic treatment and that (2) the isotopic niches of tested nematode groups would largely agree with the traditional classification of feeding groups.

## MATERIALS & METHODS

The original field experiment consisted of four different agronomic treatments, each treatment was replicated three times according to a randomised plot design and the plot size was 3 m by 10 m. The study site was No. 3 field at the Bush estate, Penicuik, Midlothian, Scotland (lat. 55°51′N, long. 3°12′W). For full site and soil details, refer to *Vinten, Vivian & Howard (1992)*, *Vinten et al. (2002)*. The conventional treatment (i.e., with the use of tillage, synthetic fertilisers, pesticides and herbicides) and the organic treatment (i.e., no fertiliser, herbicides or pesticides, but with the addition of 10 t ha$^{-1}$ of farmyard manure and under-sown with clover) were established in 2007 (*Aruotore, 2009*). Plots from these two treatments were sampled in Autumn 2014 for this study, following a crop of spring barley (*Hordeum vulgare* L.).

From each plot, 12 soil cores, 2 cm diameter and 10 cm deep, were extracted using an auger in a stratified random sampling pattern to form a composite sample. Soil samples were stored in plastic bags at 4 °C and nematodes were extracted from approximately 100 g of soil according to *Whitehead & Hemming (1965)*. The nematodes were collected alive in water every day for 16 days and kept in water at 4 °C before being identified. Each sample was examined using an inverted microscope at up to ×400 magnification. This allowed nematodes to be identified to family/genus level according to mouth and body morphology using *Bongers (1988)*. They were then transferred individually, using the microscope and an eyelash attached to the tip of an entomological needle via parafilm, into previously weighed, miniature tin capsules (8 mm × 5 mm, Elemental Microanalysis Ltd.). Additional specimens (for each group), one from every five nematodes identified, were preserved in DESS (dimethyl sulphoxide, disodium EDTA and saturated NaCl) (*Yoder & Ley, 2006*) for confirmatory identification. Tin cups with nematodes were placed inside a multi-well plate with cover but left un-sealed and dried at 37 °C overnight. A conservative target of 20 $\mu$g dry weight for each nematode taxonomic group was adopted to take advantage of the $\mu$EA–IRMS technique (*Langel & Dyckmans, 2014*).

The samples were weighed on a microbalance (Mettler Toledo) to verify if the target weight was reached. If not, more nematodes were counted into the previous day's samples, dried again at 37 °C for 12–24 h, and the process continued until the target weight was reached. Tin capsules were then wrapped and placed in a new, clean multi-well plate and shipped for measurement. Some samples that did not reach the target weight were also included for analysis.

Measurements of isotope ratios ($^{13}C/^{12}C$ and $^{15}N/^{14}N$) were made with an isotope ratio mass spectrometer (Delta V; Thermo Scientific, Bremen, Germany) coupled to a modified elemental analyser (Eurovector, Milano, Italy) as described by *Langel & Dyckmans (2014)*. Results are expressed in mUr(‰) notation after *Brand & Coplen (2012)*. SD of the system was <1 mUr at sample size of 0.6 μg N (*Langel & Dyckmans, 2014*).

Blank correction was performed by measuring additional reference samples of acetanilide ($\delta^{13}C = -29.6$ mUr, $\delta^{15}N = -1.6$ mUr) and wild boar liver ($\delta^{13}C = -17.3$ mUr, $\delta^{15}N = 7.2$ mUr). The results were used to determine the blank amount and isotopic compositions for both C and N in a Keeling-plot type graph as described e.g., in *Langel & Dyckmans (2014)*. The C blank was 2 μg with an isotopic value of −25 mUr, whereas no blank correction was performed for N because N blank was very small (0.2 μg) and variable in isotopic composition. This variability is probably caused by the fact that N is derived from two different sources, atmospheric $N_2$, on the one hand, (leading to slightly negative isotopic values due to fractionation upon diffusion) and the carryover from preceding samples, on the other hand, which can have different isotopic composition in the oxidation reactor.

All statistics and graphics were generated in R The Siber package within SIAR—Stable isotope analysis in R (*Jackson et al., 2011*) was used to analyse isotope data with Bayesian statistics. The trophic niches of the sampled nematode communities and groups were inferred from the 'isotopic niche space' occupied by each of the groups on a $\delta^{13}C/\delta^{15}N$ biplot and calculated as the Bayesian standard ellipse areas (SEA with units of $mUr^2$). In communities, the Bayesian standard ellipse areas (SEA) were probability tested to see if they were significantly different as well as comparing area overlap. Due to the small and varied sample numbers for pooled nematodes groups, area overlap of SEAs and convex hulls (TAs) were compared, both of which indicate niche width. Note that convex hull total area (TA) estimates are less reliable due to small sample sizes (*Jackson et al., 2011*), while SEA, and expressly sample size corrected standard ellipse areas (SEAc), are less biased when there are low sample numbers (*Syväranta et al., 2013*). Bayesian estimates of $10^5$ were used to generate Standard Ellipse areas in all cases.

Animals used in this research (phylum Nematoda) are not endangered, nor subject to animal research ethics regulations in the countries where the work was conducted. Field studies did not require approval by an Institutional Review Board.

## RESULTS

### Sample sizes and measurement issues

The average number of nematodes per sample (Table 1) varied within family/genus groups, some being larger in size/weight and also within samples, since both mature and
**Table 1  Nematode feeding group numbers in agronomic treatments.** The mean number of nematodes (±SD) used to achieve the target weight per sample for the groups listed, number of measured replicate samples (in brackets), and total number of measured replicate samples in each feeding group (in final column) from conventional and organic arable soils.

| | Soil nematode taxa | | Conventional | Organic | Total |
|---|---|---|---|---|---|
| Feeding group ORDER | Family | Genus | Mean no. of nematodes per sample ± SD ($n$ = measured samples) | | Number of measured samples |
| **Predators** | | | | | |
| MONOCHIDA | Anatonchidae | *Anatonchus* | – | 3 ($n = 1$) | |
| MONOCHIDA | Mononchidae | *Mononchus* | 50 ± 5 ($n = 3$) | 25.2 ± 7 ($n = 4$) | $n = 8$ |
| **Omnivores** | | | | | |
| DORYLAIMIDA | Aporcelaimidae | – | 16 ± 2 ($n = 3$) | 20 ± 3 ($n = 6$) | |
| DORYLAIMIDA | Qudsianematidae | – | – | 33 ± 4 ($n = 2$) | $n = 11$ |
| **Bacterial feeders** | | | | | |
| PLECTIDA | Plectidae | *Plectus* | 73 ± 46 ($n = 2$) | 65 ± 37 ($n = 4$) | |
| RHABDITIDA | Rhabditidae | *Rhabditis* | 32 ± 33($n = 3$) | 35 ± 14($n = 3$) | $n = 12$ |
| **Plant feeder** | | | | | |
| TYLENCHIDA | Hoplolaimidae | *Rotylenchus* | 97 ± 12 ($n = 3$) | 84 ± 27 ($n = 5$) | $n = 8$ |

immature (smaller) individuals were used, once identifiable. In the pooled samples, a priori designation of feeding type by morphology was assigned before analysis and groups included either one or two members (Table 1). Larger-sized omnivore nematodes had ranges as low as 15–25 individuals per sample, while the smaller bacterial feeders had higher ranges of 35–115 individuals to achieve 20 µg target dry weight.

For an initial quality control and check of linearity, all $\delta^{13}$C and $\delta^{15}$N (mUr) sample results were plotted against the mass of C and N per sample, respectively (Figs. 1A and 1B). Two samples (out of 39 pooled samples measured) were excluded because the C mass was considered too small. There was no significant correlation (Spearman's) between C mass and $\delta^{13}$C values ($r_s = -0.143$, $p = 0.397$), or N mass and $\delta^{15}$N values ($r_s = -0.274$, $p = 0.10$), once these two samples were removed. Importantly, there was no obvious pattern of systematic sample mass differences explaining isotopic clustering of nematode groups (Figs. 1A and 1B).

### Agronomic system comparison

The $\delta^{15}$N values for all nematode samples ranged from 1.08 to 12.79, spanning >11.5 units. When examined separately using a multivariate normality test, the conventional ($W = 0.901$, $p = 0.163$) and organic ($W = 0.940$, $p = 0.1484$) treatment groups had normal distributions. Their $\delta^{15}$N values ranged from 1.08 mUr to 12.09 mUr in the conventional treatment ($n = 12$) and from 1.99 mUr to 12.79 mUr in the organic treatment ($n = 25$).

The sample size corrected standard ellipse area (SEAc) of the conventional treatment was 11.51 mUr$^2$, while for the organic treatment it was 10.98 mUr$^2$. Bayesian generated estimates exhibited a large area overlap (Figs. 2A and 2B) between the two treatment groups, suggesting no significant difference between the size of the two SEA treatment areas ($p = 0.4928$). The standard ellipse area overlap from conventional to organic was

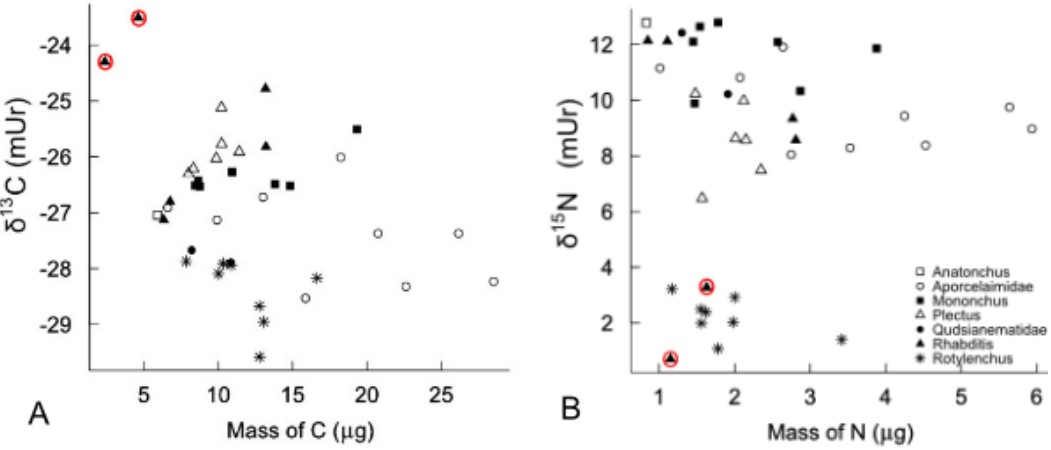

**Figure 1** (A) Sample mass of C for all samples plotted against the measured $\delta^{13}$C values. (B) Sample mass of N for all samples plotted against the measured $\delta^{15}$N values. Two samples (in red circles) were excluded as outliers.

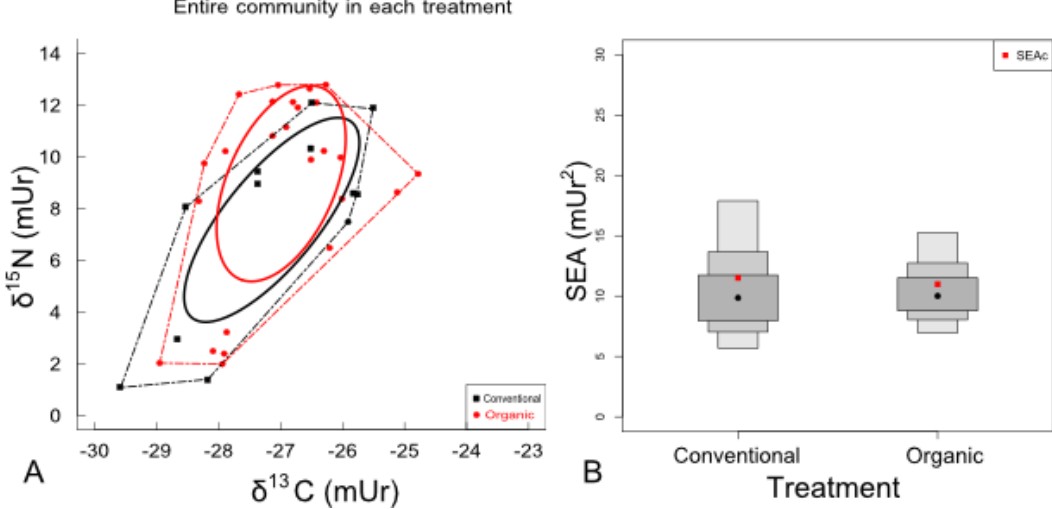

**Figure 2** (A) Biplot of N and C stable isotope ratios and (B) SIAR density plots for treatment communities. (A) All samples in the conventional agronomic treatment (black squares, $n = 12$ pooled samples) and all samples in the organic agronomic treatment (red circles, $n = 25$). The solid lines represent the Bayesian generated, Standard Ellipse area (SEAc—40% of the data) and the broken line represent the Convex Hull with 100% of the data. (B) SIAR density plot, with credible intervals (50% inside dark grey boxes, 75% middle grey boxes, 100% outer light grey boxes), for the Bayesian generated ellipses (SEA) (black dots) of the nematode isotope data overlaid with sample size corrected uncertainty around the estimates (SEAc) (red dots).

69.8% and the convex hull area overlap was 85.3%. In addition, analysis of variance showed no significant difference in $\delta^{15}$N ($p = 0.290$) or $\delta^{13}$C ($p = 0.706$) between the two treatments. Since there were no significant differences in any isotopic statistics between the two agronomic treatments, all data were pooled for subsequent feeding group analyses.

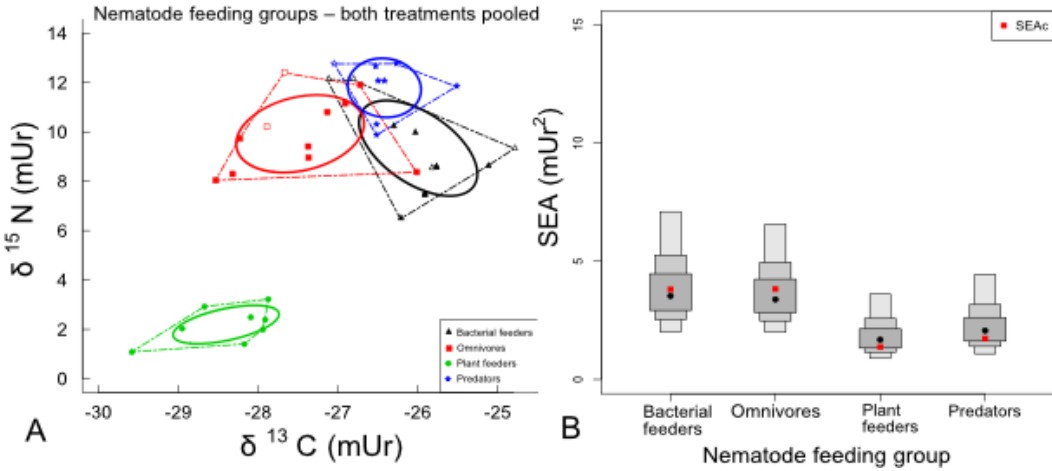

**Figure 3** **(A) Biplot of N and C stable isotope ratios and (B) SIAR density plots for nematode feeding groups.** (A) Biplot showing $\delta^{13}$C and $\delta^{15}$N of soil nematodes with Standard Ellipses (solid curved lines) and Convex Hulls (dashed straight lines) for four feeding groups: bacterial feeders (*Plectus* (solid black triangles) and *Rhabditis* (open black triangles)) ($n = 10$ pooled samples), Omnivores (Aporcelaimidae (solid red squares) and Qudsianematidae (open red squares)) ($n = 11$ pooled samples), Plant feeder (*Rotylenchus* (solid green circles)) ($n = 8$) and Predators (*Mononchus* (solid blue stars) and *Anatonchus* (open blue star)) ($n = 8$ pooled samples). (B) SIAR Density plots of Standard Ellipses areas (black dots) for the four groups with credible intervals (50% inside dark grey boxes, 75% middle grey boxes, 100% outer light grey boxes), overlaid with sample size corrected SEAc (red dots).

## Nematode feeding groups

When all samples were assigned into four groups by feeding type (Table 1), analysis of variance showed highly significant differences in $\delta^{15}$N ($p < 0.0001$) between the plant feeder and other feeders and in $\delta^{13}$C ($F_{3;33} = 24.18$ $p < 0.0001$) between all groups. The four groups (bacterial feeders ($n = 10$), omnivores ($n = 11$), plant feeder ($n = 8$) and predators ($n = 8$)) were assembled from pooled individuals from the two treatments and also from one or two different genera/families (Table 1) but with similar assumed feeding. These groups individually showed mutivariate normal distributions.

Data are graphed on a biplot ($\delta^{13}$C and $\delta^{15}$N) in 'isotopic niche space' (Fig. 3A). A significant difference in N and C stable isotope ratios between the plant feeder (*Rotylenchus*) and all other groups is apparent (Figs. 3A and 3B). The plant feeder had $\delta^{15}$N values between 1.08 and 3.22 mUr, while the predators were between 9.89 and 12.79 mUr, showing an average gap of 9.62 mUr in $\delta^{15}$N. Average C isotope ratios were also more positive (by 1.99 mUr) for the predator group ($-27.04$ to $-25.51$ mUr) compared to the plant feeder ($-29.58$ to $-27.87$ mUr). The omnivorous group had $\delta^{13}$C ($-28.53$ to $-26.01$ mUr) and $\delta^{15}$N value ranges (8.05 to 12.42 mUr) between that of the plant feeder and predators, but were elevated in $\delta^{15}$N (a difference of 7.75 mUr) compared to the plant feeder. The bacterial feeding group had a $\delta^{15}$N value range of 6.48 to 12.14 mUr and $\delta^{13}$C range of $-27.13$ to $-24.78$ mUr.

The sample size corrected Standard Ellipse Area (SEAc), representing 'trophic niche width', and Convex Hull total area (TA) were largest for omnivores (respectively 3.83 and 6.9 mUr$^2$), while the plant feeder had the smallest (1.37, 1.96 mUr$^2$) (Tables 2 and 3).

**Table 2 SEA—Bayesian generated Standard Ellipse Areas (SEAc 40% of the data, in mUr$^2$), with area and percentage overlaps.** BF, Bacterial feeders and PF, Plant feeder. 1 and 2 in parentheses represent, respectively, the first and second feeding group mentioned in the first column of the table.

| Feeding group (1) & (2) | Area (1) | Area (2) | Area overlap | % overlap |
|---|---|---|---|---|
| PF & predators | 1.37 | 1.73 | 0 | 0 |
| Omnivores & PF | 3.83 | 1.37 | 0 | 0 |
| BF & PF | 3.81 | 1.37 | 0 | 0 |
| Omnivores & predators | 3.83 | 1.73 | 0 | 0 |
| BF & omnivores | 3.81 | 3.83 | 0.037 | <1% |
| BF & predators | 3.81 | 1.73 | 0.31 | 8–18% |

**Table 3 Convex Hull (100% of the data, in mUr$^2$) with area and percentage overlaps.** BF, Bacterial feeders and PF, Plant feeder. 1 and 2 in parentheses represent, respectively, the first and second feeding group mentioned in the first column of the table.

| Feeding group (1) & (2) | Area (1) | Area (2) | Area overlap | % overlap |
|---|---|---|---|---|
| PF & predators | 1.96 | 2.33 | 0 | 0 |
| Omnivores & PF | 6.94 | 1.96 | 0 | 0 |
| BF & PF | 5.82 | 1.96 | 0 | 0 |
| Omnivores & predators | 6.94 | 2.33 | 0.34 | 5–15% |
| BF & omnivores | 5.82 | 6.94 | 1.61 | 23–28% |
| BF & predators | 5.82 | 2.33 | 0.90 | 15–38% |

Predator SEAc and TA were also small (1.73, 2.33 mUr$^2$). The SEAc or TA of the plant feeder did not overlap with any of the other groups. There was some TA overlap between the bacterial feeders and the omnivores (23–28%) and between the bacterial feeders and predators (15–38%), but minimal overlap between the omnivores and predators (5–15%) (see Table 3). There was no significant overlap in SEAc's between bacterial feeders and omnivores (1%), however they were in the same $\delta^{15}$N range (representing trophic level) and there was a small SEAc overlap between bacterial feeders and predators (<8–18%).

## DISCUSSION

### Sample sizes and measurement issues

The near-conventional $\mu$EA–IRMS technique allows the use of microgram samples, reducing the time-consuming effort for enumerating nematode groups experienced by *Moens, Bouillon & Gallucci (2005)* and others. Nematodes from four feeding groups were included in this study. Fungal feeders were omitted because of their small body size (hence practically unattainable numbers required to reach target weight), low abundances and the difficulty in identifying live specimens at the required taxonomic resolution. The numbers necessary to reach the sample weight for conventional isotopic analysis are difficult to achieve, especially by the approach used here. For example, because of this difficulty, *Kudrin, Tsurikov & Tiunov (2015)* used nematode sample weights as low as 11 $\mu$g despite using conventional IRMS for isotope analysis. Bayesian community metrics,
more conservative methods than convex hull area, were used for inference of trophic behaviour to redress the limitations of small sample numbers.

## Nematode feeding groups

Prior studies have used isotopic analysis to decode nematode contribution to soil food webs but none has attempted to test members of the traditional soil nematode feeding groups composed by *Yeates et al. (1993)*. To this end, the present study somewhat parallels that of *Kudrin, Tsurikov & Tiunov (2015)* on one forest soil in Russia, with the exception of the use of the $\mu$EA–IRMS method, the inclusion of two arable treatments and the successful analysis of a plant-feeding group. Based on dual C and N natural isotope abundance measurements of members of the soil nematode community, results from *Kudrin, Tsurikov & Tiunov (2015)* and the present study conform to (independently of each other) major aspects of the widely used feeding group concept. For the most part, there is agreement between isotopic and traditional feeding groups emerging from both these studies, largely agreeing with morphology-based categorisation to feeding groups. However, isotopic compositions indicate that some members diverge from assumed feeding, which is further discussed below. Many of the uncertainties discussed here may be caused by pooling of species and higher taxa, and these uncertainties will be resolved in future studies that measure better delineated genera or even species of soil nematodes. Life stage of individuals may also be taken into account.

### Plant feeders

Soil food webs are characterised by two distinct resources, living plant roots and detritus (*De Ruiter et al., 1993*), with the majority of soil groups consuming from the detrital food web (*Korobushkin, Gongalsky & Tiunov, 2014*). The $\delta^{15}$N of non-plant feeders, namely, saprophagous omnivores, bacterial feeders and fungal feeders, in soil food webs are elevated through the assimilation of microbially-processed organic matter with a marked isotopic distance from plant matter (*Hendrix et al., 1999*). In addition, predators are distant from primary plant resources via consumption of $\delta^{15}$N-elevated prey. A resource distinction is clearly evident in the nematode data between the assumed plant feeder and all other groups (Fig. 3A).

Plant feeders ostensibly have the same or slightly enriched $\delta^{15}$N values as their resources, and depleted C and N isotope ratios compared with other soil fauna usually reflect feeding on plants or fresh plant residues (*Schmidt et al., 2004*; *Illig et al., 2005*; *Maraun et al., 2011*), as displayed by *Rotylenchus* in this study. Here, what is most apparent is a distinct dual trophic grouping, encompassing predators, omnivores and bacterial feeders presumably feeding on detritivore resources and another grouping with the plant feeder directly consuming plant roots. *Rotylenchus* was depleted in both $^{15}$N and $^{13}$C compared to all other groups suggesting that categorization of the group as plant feeding is correct.

The plant feeder had the smallest SEAc, reflecting a narrow niche width with a singular food source, with their role as direct plant feeding. This may change seasonally due to changing plant nutrient supply (*Cesarz et al., 2013*) or be affected by the management of the crop in an arable system. As only one genus is represented here, it cannot be inferred that this will be the case for all plant feeders.

### Predators

At the other extreme, the predatory group (mainly *Mononchus*) had the most elevated $\delta^{15}$N of the nematode groups, as is common for predators in soil food web studies where they are at the top of the food web and are relatively $^{15}$N enriched in relation to their diet (*Scheu & Falca, 2000*; *Maraun et al., 2011*). The isotopic $\delta^{15}$N distance between predators and omnivores or bacterivores does not clearly indicate a full step in trophic level between these three groups, but the $\delta^{15}$N spacing between the plant feeder and predators suggests an apparent difference of 3–4 trophic levels within the soil nematodes tested. This distance might indicate that predators have a feeding preference for prey from higher trophic levels than plant feeders. As such, the predators likely feed more on other predators, omnivores and bacterial feeders (and presumably fungal feeders) and less so on plant feeders.

Predatory feeders displayed a small SEAc, suggesting that their diet is not general but specific to feeding on small, higher trophic level soil animals, reflected by their elevated $\delta^{15}$N values (9.89–12.79 mUr). This feeding presumably involves intraguild predation (*Illig et al., 2005*), by contrast if the plant feeder ($\delta^{15}$N 1.08–3.22 mUr) was being consumed, the values would have been expected to be lower. On the other hand, predator $\delta^{15}$N was expected to be markedly more enriched than that of bacterial feeders. Consumption of plant feeders by predators could be one explanation for this. Also, the more negative $\delta^{13}$C of predators compared to bacterial feeders could be explained by biochemical differences rather than feeding habits, for example predators could have larger lipid reserves that are more negative in $\delta^{13}$C compared to proteins and carbohydrates (*Schmidt et al., 2004*). It must also be noted that here mainly one genus, *Mononchus*, is represented. As both mature and immature specimens were used, life stage feeding may be a factor affecting the isotopic composition of the group i.e., immature Monochidae are thought to feed on bacteria (*Yeates, 1987*).

### Omnivores

Omnivores had a larger SEAc (isotopic niche width) suggesting a wider trophic niche and thus assimilation of a variety of resources, adhering to their definition in nematology as generalist feeders. This reflects the feeding by omnivores reviewed by *McSorley (2012)* and assumed by *Yeates et al. (1993)* who described omnivores as feeding widely on fungal, deposit, bacterial and predatory reserves from non-nematode and nematode sources. Using the biplot and Convex hull (Table 3) overlaps between omnivores and bacterial feeders, there is a suggestion that omnivores and bacterivores occupy the same trophic level (second highest). This is at odds with *Kudrin, Tsurikov & Tiunov (2015)*, where the omnivores and predators appear to share the highest trophic level. This could be explained by different members representing the omnivore families from the two studies or by different behaviour in different habitats.

The overall sequence of groups (bacterial feeders, omnivores and predators) on the $\delta^{13}$C and $\delta^{15}$N bi-plot and therefore in 'trophic niche space', in this arable study corresponds somewhat with that of the *Kudrin, Tsurikov & Tiunov (2015)* study, from a taiga spruce forest soil but is not the same. The SEAc and TA overlaps of these three feeding groups might support the theory that 'true' omnivory is more prevalent in other than just omnivores (*Moens, Traunspurger & Bergtold, 2006*).

### Bacterial feeders

Not all a priori groupings, based on morphology, clearly fit to *Yeates et al.*'s (*1993*) feeding categorisation. The SEAc of bacterial feeders was comparatively large and they had isotopic values that were somewhat ambiguous with a small degree of 'trophic niche' overlap with predators. The bacterial feeders were more $^{15}$N and $^{13}$C enriched than expected. Two genera were represented in the group. Diverse feeding between the two genera may have influenced the size of the SEAc as well as the overlap. Bacterivores $^{13}$C enriched could reflect grazing on bacteria that are colonizing older elevated $^{13}$C food resources in soil (*Schmidt et al., 2004*) and were $^{15}$N enriched which could suggest some predatory behaviour like aquatic deposit feeding nematodes in the study by *Moens, Bouillon & Gallucci (2005)*. Present samples were taken from post harvest soils where there were fewer inputs from a growing crop, so older carbon may be accessed from bacteria colonizing plant residues, applied manure and soil organic carbon with elevated $^{15}$N as shown by *Scheunemann, Scheu & Butenschoen (2010)*. Bacterivores could also acquire elevated $\delta^{15}$N values by feeding on bacteria fuelled by livestock manures that can be highly $^{15}$N enriched due to gaseous losses of isotopically light N during storage (*Schmidt & Ostle, 1999*). The bacterial feeder/predator overlap could also be accounted for by direct microbial feeding by predators (*Wardle & Yeates, 1993*).

The overlap with predators may also be due to a lower than expected N fractionation. More information is becoming available on trophic distances between feeding groups in soil food webs, as evinced by a recent stable isotope meta-analysis (*Korobushkin, Gongalsky & Tiunov, 2014*)., However, the 'trophic distance' in soils is less clear than between trophic levels (i.e., 3.4 mUr for $\delta^{15}$N) in other systems (*Hendrix et al., 1999*), with soil food webs having more trophic levels than other food webs (*Digel et al., 2014*). In addition, the underlying body-diet spacing of consumers are poorly documented and can be affected by the type of trophic level, feeding guilds within feeding groups, or by environmental or physiological factors (*Schneider et al., 2004*; *Maraun et al., 2011*). For instance, a meta-analysis suggested that the $^{15}$N enrichment can be higher in detritivores and lower in herbivores relative to their food source, and that the type of N excretion of different taxa can have an influence on trophic distance (*Vanderklift & Ponsard, 2003*). *Moens et al. (2014)*, however, observed spacings as high as $\geq$4 mUr between microalgae and nematode grazers in soft sediments.

## Agronomic system comparison

The hypothesis that the nematode feeding ecology reflected by isotopic data would show a difference between conventional and organic agronomic treatments was not supported. Organic systems have been shown to cause a shift in trophic responses compared with conventional (*Haubert et al., 2009*; *Sánchez-Moreno et al., 2009*), for instance because external carbon inputs such as manure strongly influence the energy pathway in soil food webs (*Crotty et al., 2014*). In agricultural soils, management and resource availability have a large influence on the resulting energy pathway (*Zhao & Neher, 2014*). The energy pathway (plant, bacterial or fungal based, see *Neher (2010)*) in a detrital consumer soil system can influence the number of trophic levels (*Illig et al., 2005*). However, found little difference in nematode maturity and trophic diversity indices from organic to conventional cropped

fields. Similarly, in the present study the agronomic treatments did not vary significantly, which could reflect the time lag before management changes have an effect on the soil system or the fact that baseline food resources in the two systems were essentially the same.

## Applications for soil ecology

The present work is in line with prior studies and upholds many long held assumptions of trophic behaviour of members of certain nematode feeding groups. By using the μEA–IRMS technique, it is now possible to confirm on a scale as fine as species level (for larger species at least) the feeding behaviour of identifiable soil nematodes. This will further highlight nematode feeding and their role in the complexity of the wider soil food web. Such is the power of isotopic techniques for trophic inference, future studies may find terrestrial genera/species that clearly do not fit the assumed morphological and ecological feeding previously assigned to them, as was the case in aquatic studies (*Moens, Bouillon & Gallucci, 2005*; *Estifanos, Traunspurger & Peters, 2013*; *Vafeiadou et al., 2014*). Considering the close relationship between terrestrial and aquatic nematode feeding groups, the present work also has relevance to the feeding ecology of aquatic nematodes.

One unique feature of the soil food web is the co-existence of many decomposer groups (*Illig et al., 2005*). Year round active nematodes encompass many of the wide range of feeding types found within the soil food web and as such are an excellent soil bioindicator group (*Ferris, Bongers & De Goede, 2001*; *Ferris et al., 2012*; *Ritz & Trudgill, 1999*; *Neher, 2010*). Trophic information can help to identify 'sentinel' nematode taxa that reflect aspects of soil ecosystem function on landscape monitoring scales (*Neher, 2010*). Isotope techniques can be used to look at temporal changes in nematode feeding in response to different ecological contexts or management, such as pollution monitoring and habitat restoration (*Neher, 2010*) or climate change (*Sticht et al., 2009*).

The validity of morphology (mouthparts) linking form to function (*Ritz & Trudgill, 1999*) is confirmed here by isotopic analysis on certain nematodes. Although many taxa have yet to be tested, feeding group members were isotopically confirmed by *Kudrin, Tsurikov & Tiunov (2015)* as well as the present study, further substantiating the effectiveness of nematode indices based on feeding strategies. The small sample sizes needed for trophic analysis and demonstrated here could complement functional food web detail at a genus/species level that is usually lacking from guild-based indices systems.

Species level isotopic investigations of soil nematodes can resolve many of the uncertainties discussed here caused by pooling of species or higher taxa. For quantitative studies, the same analytical approach used here could be combined with isotopic labelling of plants or other food sources (e.g., *Crotty et al., 2014*; *Schmidt, Dyckmans & Schrader, 2016*; *Shaw et al., 2016*). Such studies can estimate the flow of C and N from resources (e.g., bacteria, algae, plant roots) to nematode taxa, but at a finer taxonomic resolution. This would offer a better understanding of the feeding ecology of nematodes and their trophic interactions in soil food webs.

## ACKNOWLEDGEMENTS

We appreciate the extensive and helpful feedback of reviewer Tom Moens. We thank Walter S. Andriuzzi for reading and commenting on the paper and Bernard Kaye for advice on graphics software.

### Funding

This work was funded by the Earth and Natural Sciences (ENS) Doctoral Studies Programme. The ENS programme is funded by the Higher Education Authority (HEA) through the Programme for Research at Third Level Education, Cycle 5 (PRTLI-5) and is co-funded by the European Regional Development Fund (ERDF). SRUC is partly funded by the Scottish Government's Rural and Environment Science and Analytical Services Division. The funders had no role in study design, data collection and analysis, decision to publish, or preparation of the manuscript.

### Grant Disclosures

The following grant information was disclosed by the authors:
Earth and Natural Sciences (ENS) Doctoral Studies Programme.
Higher Education Authority (HEA) through the Programme for Research at Third Level Education, Cycle 5 (PRTLI-5).
European Regional Development Fund (ERDF).
Scottish Government's Rural and Environment Science and Analytical Services Division.

### Competing Interests

The authors declare there are no competing interests.

### Author Contributions

- Carol Melody conceived and designed the experiments, performed the experiments, analyzed the data, wrote the paper, prepared figures and/or tables, reviewed drafts of the paper.
- Bryan Griffiths contributed reagents/materials/analysis tools, reviewed drafts of the paper, confirmation of nematode identification.
- Jens Dyckmans contributed reagents/materials/analysis tools, reviewed drafts of the paper.
- Olaf Schmidt conceived and designed the experiments, analyzed the data, wrote the paper, reviewed drafts of the paper.

### Data Availability

  The raw data can be found in Data S1.

### Supplemental Information

Supplemental information for this article can be found online at http://dx.doi.org/10.7717/peerj.2372#supplemental-information.

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
