# Peer review of "Stable isotope analysis (δ13C and δ15N) of soil nematodes from four feeding groups"

_PeerJ, doi:10.7717/peerj.2372_

## Round 0.1 · original submission · Major Revisions

The paper has been reviewed by 2 experts in the field. Please revise and take into considerations the suggestions, especially Reviewer 2 on the interpretation of data and conclusions drawn from them

Reviewer 1 ·

Basic reporting

Key words are provided by the authors, but PeerJ does not make any comments how to list key words.
No further comments.

Experimental design

No Comments

Validity of the findings

No Comments

Additional comments

Introduction
L51: I suggest to name here all trophic groups you considered in your study and not just list plant feeders and predators.
L68: You may add here also that responses within trophic groups can be reverse due to different life stage histories within one trophic group, see Neher 2013 (Journal of Nematology) and Cesarz et al. 2015 (Pedobiologia)
L85: Please explain mUr, this should be per mill, right? Later in the text you use mUr². What is the difference?
L90-98: I am not sure about the purpose of this paragraph. For instance, what does “were slow to follow” mean. Especially the given example does not help to understand what you want to point out here.
L102: is established the right word? I suggest to use “showed” or the like

Material and Methods
L143: Can you please add information about how you selected single nematodes from your samples? I know this is not easy and you may provide an effective technique?!
L145: What does (for each group, 1 nematode in every 5) mean?
L160: This unit is new to me but I am happy to get the reference here now. However, I do not understand why you state that results are expressed in mUr² and in the following you leave out the ² (exponent two)

Results
Table 1: Please format your table according to scientific standard, e.g. never use vertical lines
L183: Please remove those square symbols which appear in the pdf version.
L197: I suggest to remove R code from text
L213/Table 2: What is SIAR? It is mentioned here first, you should explain this.
L219: Please write numbers up to twelve as one, two, three etc
L222: It is highly appreciated that you showed degrees of freedom. I would like to see those also for the remaining statistics, e.g. L220. Please be consistent.
Table 2 & 3: Please explain in the header what (1) and (2)/Area (1)/Area (2) mean. Further, remove vertical lines.

Discussion
L256: 4 = four
L265: Can you please provide more qualitative information by citing the work of Kudrin et al? At the moment it just says that there is another study but you should invest more effort to highlight the differences/advantages of your/the other study
L283: Please do not use “too”. You may write: In addition, ….
L307-310: And this suggests ….
L316: By using the word ambiguous you should tell the reader what you expected/did not expect
L323: It is known that juvenile Mononchidae feed on bacterial prey before they switch their diet to real prey
L329: Can you please explain why they should feed on small slow-moving organisms
L334: Please note that in the pdf version the µ is replaced by a square
L353: time lag

·

Basic reporting

Most of my comments relate to this aspect, but I have listed everything under 'general comments'.

Experimental design

No comments, apart from (see general comments) that only one or two genera per feeding type do not warrant very general statements about the validity of feeding type classifications.

Validity of the findings

Findings are valid, interpretations not always very OK. See under general comments.

Additional comments

The study by Melody et al. presents very interesting and useful data and a methodological advancement. It deserves publication in a good journal. However, I do have several issues with the manuscript as it currently stands. Several of these issues follow from – in my opinion – insufficiently careful interpretation of data. I try to clarify these below, along with one methodological and several minor comments.
I can probably best organize my main concerns by ‘tackling’ the title of the manuscript.
First, this is not a micro-scale analysis (when I read that title, I expected to read about microspatial patchiness or so), but an analysis with genus-level resolution. That, of course, is a purely semantic comment.
However, I also disagree more fundamentally with both the words ‘independent’ and ‘confirmation’. The confirmation is not ‘independent’, because it uses a priori defined groupings of nematodes according to established feeding types to analyse the data. While this may again seem a semantic issue, it most certainly is not. I illustrate this by referring to figure 3. One major conclusion here is that the SEAc’s of omnivores are larger than those of other feeding types. However, just assume for a moment that you had not a priori qualified your nematodes in feeding types based on morphology, or that you would have had a reason to assign one of the two omnivore species as a predator or even a bacterivore, then your isotopic results would have found a smaller SEAc for omnivores than for predators or bacterivores. Similarly, only some data points clearly indicate separate trophic levels. Without your a priori knowledge on feeding types, you most likely would have concluded on a different clustering of nematode genera. So your data are in line with expectations based on morphology-based classifications, but they do not provide independent evidence.
Being in line is also not the same as really confirming. Please bear in mind that one of your feeding types is represented by a single species here, and the others by two species each. Although this is clearly mentioned, the message of the paper still reads like a proof that traditional feeding type classifications hold. But conclusions could have been quite different had other species been included. Moreover, the very close proximity between some of the bacterivore samples and several predator and omnivore samples does not confirm that bacterivores are simply bacterivores, nor that predators are predators or omnivores are omnivores. What this actually does suggest is that omnivory may be more rule than exception in soil nematode feeding: bacterivores may also prey on, for instance, heterotrophic protists, which renders them omnivores in the true ecological sense of the word (organisms feeding from more than one trophic level), and/or predators may feed on bacteria or microalgae in the soil, again implying omnivory. Incidentally, in nematology the term omnivores is used inaccurately as a semi-synonym for generalist feeders (see Moens et al. (2004) in Nematology Monographs and Perspectives 2). It would be good if the authors could clarify this early on in their ms, so as to avoid confusion among a non-nematological readership.
I would therefore suggest to modify the title into something like (not well though of, you can undoubtedly come up with something better) ‘Genus-level dual stable isotope analysis is in line with expectations based on traditional feeding-type classifications’.
I’d like to continue for a moment on the explanations given for the close clustering of omnivores, bacterivores and predators in fig. 3. While most of the possible explanations are given in the discussion, I felt they were not always sufficiently clearly explained. In my opinion, the small discrepancy in delta15N between bacterivores and predators can result from a) a lower than expected N fractionation. Unlike stated in lines 287-288, trophic distances are not very clear nor consistent in any habitat. Such differences may differ between trophic levels (TL) and may even change according to environmental and physiological factors. However, if we imagine a short food chain from detritus over bacteria and bacterivores to predators, we could be tempted to agree with some paper(s) (I do not remember the correct references by heart) suggesting that trophic fractionation tends to be smaller at lower TL. Then again, Moens et al. (2014) found a fractionation ≥ 4 mUr between microalgae and their nematode grazers. This deserves some more focused discussion here. An alternative option is that, as mentioned by the authors, b) the predators also consume bacteria and/or the bacterivores also prey on bacterivores protists. A final option, which is hinted at, but insufficiently explained, is that c) manure is the main fuel for bacteria in these agrosystems and has elevated delta15N, resulting in ‘artificially’ elevated delta15N of bacterivores, much like sewage sludge in estuaries tends to be enriched in 15N. I am not sufficiently aware of agricultural systems to know if this would also apply to manure. In any case, none of the above options satisfactorily explains why the delta13C of bacterivores is slightly enriched compared to that of predators and omnivores. This is counterintuitive, unless predators (and omnivores) do not only feed on bacterivorous prey and on detritus chains, but also on other prey and perhaps other chains. It is possible that predators and omnivores do have some herbivores prey as well, which would lower both their delta13C and – even more so – their delta15N. The authors may argue that many of these points are somehow addressed in their discussion, but this discussion should be better structured and the authors should better highlight the different options and their implications.
Given my earlier comment about the paper giving the impression of a broad support of feeding-type classifications based on only one or two genera per feeding type, I’d like a) the authors to modify figure 3a so that symbols visually clarify which data point corresponds to which genus. For instance, the distinction between omnivores and predators is not all that evident (for some marine nematodes, species with very well documented voracious predatory behaviour now appear to be real omnivores and generalists). So I would like to be able to pinpoint data points to genera in fig. 3a.
In that same figure (+ 3b), the contention that the SEAc’s of omnivores are larger than those of other groups is clear for predators and plant feeders, but not really so for bacterivores.
Lines 288-289: I feel there is a risk for confusion when talking about numbers of trophic levels at the food web level rather than at the food chain level. Fig. 3 suggests that the herbivore does not contribute much to the other feeding types. So the food chain complexity (= number of TL) may be quite limited after all, despite your food web covering more TL because it comprises different chains which may be largely separated. There is no way to prove that with only one herbivore analysed, but there is also no way to suggest here that trophic complexity would be quite high.
The suggestion at lines 311 and further than omnivores had a larger SEAc than other feeding types again sounds partly misleading when it is used to confirm the idea that these omnivores have a broader diet than the other feeding types. This larger SEAc could simply result from the choice of (only two) genera for the analysis. If these happen to be more different in their trophic strategy than the two predators and the two bacterivores, then you get this result. But it does not prove omnivory or even generalism. To do so, one should look at the SEAc at the level of individual species/genera, or of many genera per trophic group, which understandably has not been done here.
I would like to re-iterate the issue of the word ‘confirm’ in the title. Vafeiadou et al. (2014) demonstrated that even genera belonging to the same family could differ – not only in resource use, but even in trophic level. One cannot accuse the authors of not mentioning such cautions, but to the general readership, this paper really reads much less nuanced, and the authors should take care to clarify the limitations of their conclusions throughout the ms.
Line 329: I do not know where the specification ‘slow-moving’ is coming from. Not from your data, in any case.
With a limited effort, the authors could expand their intro and discussion so as to include marine and freshwater nematodes. Unfortunately, many aquatic nematologists read few soil nematology papers and vice versa. It would not require much effort to highlight the relevance of this study for aquatic nematology as well.
Line 121: ‘the isotopically represented nematode communities’: I do not understand what you mean here.
My methodological concern: the authors state they have applied blank corrections according to Langel and Dyckmans (2014). Given that the latter is among the authors of the current paper, I feel fairly comfortable. But I should feel equally comfortable without that, and I would not, because Langel & Dyckmans mention several blank correction procedures. They also highlight the fact that tin cups tend to yield blanks which a) are far from negligible when working with such small sample masses, and b) vary between batches up Sn cups. I agree with both comments, based on my own experience. A) is one of the reasons why I prefer to use smaller cups, because they yield lower blanks for carbon. I would like the authors to be much more explicit about their blank correction procedure, both in determining correct isotopic signatures of blanks (see Langel & Dyckmans 2014) and in how to calculate the correction (see Moens et al. 2014).
Figure 1 shows correlations between delta values and carbon or nitrogen mass. However, the ms does not state how C and N mass were measured. It only talks about dry weight. I assume reference curves have been used here, but this should be clarified.
Lines 199-200: please also mention delta13C range.
A final comment about fig. 3: the analysis tests differences in SEAc, which is essentially differences in niche breadth. But two niches can be very distinct and yet equally large (same niche space but no or limited overlap). Is that aspect in any way included in the data analysis, and if so, how?

---

## Round 0.2 · accepted · Accept

All of the reviewers' comments have been addressed.